# Clinical Efficacy and Safety of Chinese Herbal Medicine in the Treatment of Uremic Pruritus: A Meta-Analysis of Randomized Controlled Trials

**DOI:** 10.3390/ph15101239

**Published:** 2022-10-09

**Authors:** Ping-Hsun Lu, Chien-Cheng Lai, I-Hsin Lin, Fu-Ming Tsai, Po-Hsuan Lu

**Affiliations:** 1Department of Chinese Medicine, Taipei Tzu Chi Hospital, Buddhist Tzu Chi Medical Foundation, New Taipei City 23142, Taiwan; 2School of Post-Baccalaureate Chinese Medicine, Tzu Chi University, Hualien 97048, Taiwan; 3Department of Medical Education, MacKay Memorial Hospital, Taipei 10449, Taiwan; 4Department of Research, Taipei Tzu Chi Hospital, Buddhist Tzu Chi Medical Foundation, New Taipei City 23142, Taiwan; 5Department of Dermatology, MacKay Memorial Hospital, Taipei 10449, Taiwan; 6Department of Medicine, MacKay Medical College, New Taipei City 23142, Taiwan

**Keywords:** uremic pruritus, chronic kidney disease, chronic renal failure, Chinese herbal medicine, systematic review, meta-analysis

## Abstract

Uremic pruritus is a disturbing and refractory symptom in patients with advanced chronic kidney disease. Chinese herbal medicine has been reported to alleviate uremic pruritus. To investigate the effects of Chinese herbal medicine, we conducted a systematic review and meta-analysis on patients with uremic pruritus. We searched databases (prior to 3 May 2022) for randomized controlled trials on the effects of Chinese herbal medicine in treating uremic pruritus. Our meta-analysis included 3311 patients from 50 randomized controlled trials. In patients with uremic pruritus, adjunctive Chinese herbal medicine significantly improved overall effectiveness (risk ratio 1.29, 95% CI 1.23 to 1.35), quality of life, renal function, reduced pruritus score, and inflammatory biomarkers compared to control groups with hemodialysis alone or with anti-pruritic treatments. Chinese herbal medicine treatment showed a time-dependent tendency in improving the visual analog scale of dialysis patients. Compared to control groups, no significantly higher risk of adverse events in patients taking Chinese herbal medicine (risk ratio 0.60, 95% CI 0.22 to 1.63). Chinese herbal medicine appears to be effective and safe in complementing the treatment of patients with uremic pruritus.

## 1. Introduction

Uremic pruritus (UP) or chronic kidney disease-associated pruritus is a serious and burdensome symptom of advanced chronic kidney disease [1]. The prevalence of uremic pruritus in dialysis patients is about 40% [2], but the pathophysiology of uremic pruritus remains unclear. Previous studies of patients with UP have found elevated levels of blood urea nitrogen (BUN), calcium (Ca), phosphorus (P), and parathyroid hormone (PTH) [3]. However, the relationship between electrolyte concentrations and severity of UP is controversial. High Ca concentrations have been reported in association with UP [4]. In conducting a cross-sectional study, Makhlough [5] showed that level of intact parathyroid hormone (iPTH) is correlated with severity of UP. In contrast, a randomized-controlled trial (RCT) [6] and a multicenter study [7] displayed no association in dialysis patients between pruritus severity and serum concentrations of P, iPTH, PTH, or Ca. In addition, recent studies have shown that UP is associated with inflammation, specifically in elevated levels of tumor necrosis factor-alpha (TNF-α), interleukin (IL)-2, IL-6, C-reactive protein (CRP), high-sensitivity CRP, and β2-microglobulin (MG) [3,8,9,10,11,12]. For instance, Kimmel et al., reported that CRP and IL-6 were significantly higher in dialysis patients with UP than patients without UP and that UP patients also showed a nonsignificant elevation of TNF-α [10]. Besides inflammation, UP impacts on quality of life (QOL) of patients. One study with two hundred dialysis patients showed significantly lower quality of life indices in patients with UP [13].

Systemic and topical agents, phototherapy, and alternative medicines have been reported as treatments for UP. As a recent therapeutic algorithm, complementary alternative medicine was considered as adjunctive treatment while refractory status after systemic treatments such as anticonvulsants, opioid receptor agonists, or antihistamines [14]. However, gabapentin and antihistamine have been shown to cause dizziness, drowsiness, and somnolence [15,16]. In addition, sunburn and tanning have been noted as side effects of phototherapy [1,17]. Acupuncture and topical capsaicin appear to ameliorate UP, although topical capsaicin treatment often causes burning sensations or erythema [18]. Identifying effective and safe complementary treatments for UP patients, such as Chinese herbal medicine (CHM), is desired.

Chinese herbal formula, such as uremic clearance granules (UCG), has been reported to improve renal function and lower the serum concentration of BUN, serum creatinine (SCr), PTH, iPTH, P, and inflammatory biomarkers in UP patients, without significant adverse effects [14]. In addition, there has been limited evidence regarding the effects of CHM in treating UP patients. Therefore, we conducted a systematic review and meta-analysis to evaluate the efficacy and safety of CHM in UP patients.

## 2. Results

### 2.1. Characteristics of Included Studies

We used a PRISMA flowchart to illuminate the process of identifying and selecting RCTs in evaluating the effects of CHM for UP patients (Figure 1). We identified 2145 articles from electronic databases and 18 additional records obtained through other sources. We excluded 1544 articles based on their titles and abstracts. We then reviewed the full texts of the remaining 137 articles. We excluded 87 articles due to the following reasons: 19 studies were review articles, 19 studies were not RCTs, 33 studies involved different interventions, 5 studies did not involve UP patients, 8 studies did not report data, 2 studies involved overlapping populations, and 1 study was retrospective. We qualitatively and quantitatively synthesized the remaining 50 articles.

Characteristics of the included RCTs are listed in Table 1 and Appendix A. All trials were published between 2003 and 2022, which included 3311 participants. Sample size per study ranged from 30 to 128 participants. The management of the control group and the intervention group is listed in Table 1. Most of the patients in the control group were undergoing hemodialysis. Some dialysis patients underwent high-reflux hemodialysis. Thirteen trials assessed additional treatments, including antihistamines and calamine lotion in the control group [19,20,21,22,23,24,25,26,27,28,29,30,31]. There were 4 studies including patients not undergoing dialysis [32,33,34,35]. For studies following the treatment of UP patients undergoing dialysis, 10 studies involved UCG [36,37,38,39,40,41,42,43,44,45], 6 studies assessed patients treated with Touxie-Jiedu-Zhiyang decoction [46,47,48,49,50,51], 4 studies examined patients treated with Yangxue-Runfu-Yin [52,53,54,55], and 13 studies investigated the efficacy of other CHM formulas [56,57,58,59,60,61,62,63,64,65,66,67,68]. Components of all CHM formulas are listed in Appendix A.

### 2.2. Risk of Bias

Risk of bias is presented in Figure 2. In the randomizing process, thirty-four studies mentioned the methods of randomization. Only four studies [19,27,28,34] revealed details about their allocation concealment. Five studies [20,22,23,56,60] did not describe the baseline conditions of patients or provide statistical data, both of which could result in baseline imbalances. In addition, no double blinding was performed in the reviewed trials. Three studies [27,34,54] were not intention-to-treat analyses and had more than 5% of losing outcome data. Bias of outcome measurements, including visual analog scale, itch-intensity ratings, overall effectiveness, and quality of life scale, were high based on self-assessment. Two studies [32,63] were potentially biased because they only reported a subset of their data.

### 2.3. Primary Outcome

#### 2.3.1. Pruritus Severity (Visual Analog Scale (VAS), Duo, Dirk R. Kuypers Itching Scale)

Pruritus severity was assessed by a VAS score in 21 studies (Figure 3a) [21,24,27,28,30,35,39,40,41,42,43,44,45,46,47,51,54,55,58,62,64]. When compared to controls, CHM reduced VAS scores significantly (mean difference [MD] −1.98, 95% CI −2.23 to −1.73). Depending on duration of dialysis, patients were treated with CHM, CHM was shown to significantly reduce VAS scores after <8 weeks of treatment (MD −1.68, 95% CI −2.21 to −1.16), ≥8 weeks of treatment (MD −1.74, 95% CI −2.32 to −1.17), and ≥12 weeks of treatment (MD −2.12, 95% CI −2.85 to −1.39) (Figure 3b). The longer the dialysis patients were treated with CHM, the more reduction in VAS scores was shown. Moreover, a variety of Chinese herbal formulas significantly reduced VAS scores: Touxie-Jiedu-Zhiyang decoction (MD −2.44, 95% CI −4.40 to −0.47), UCG (MD −2.05, 95% CI −2.19 to −1.92), and other decoctions (MD −1.75, 95% CI −2.16 to −1.35) (Figure 3c).

The Duo and Dirk R. Kuypers itching scales for pruritus assessment were reported by seven [26,34,35,53,54,55,63] and two studies [31,36], respectively. CHM significantly reduced pruritus symptoms according to both the Duo (MD −6.11, 95% CI −8.28 to −3.94) and Dirk R. Kuypers (MD −5.12, 95% CI −6.49 to -3.75) itch-intensity scales (Figure 4a,b). Heterogeneity across trials was high for VAS scores (I^2^ = 75%, *p* < 0.00001, VAS scores ≥8 weeks of treatment (I2 = 70%, *p* = 0.005), ≥12 weeks of treatment (I2 = 87%, *p* < 0.00001), for the Touxie-Jiedu-Zhiyang decoction (I2 = 96%, *p* < 0.00001), for other decoctions (I^2^ = 65%, *p* = 0.0002), and relative to pruritus scores using the Duo itch-intensity scale (I^2^ = 87%, *p* < 0.00001). However, heterogeneity of VAS scores (pruritus severity) was not significant after <8 weeks of treatment with CHM (I^2^ = 49%, *p* = 0.12) or with UCG (I^2^ = 0%, *p* = 0.88); similarly, pruritus scores based on the Dirk R. Kuypers itch-intensity scale was also not significant (I^2^ = 0%, *p* = 0.74).

#### 2.3.2. Overall Effectiveness

Thirty-seven studies reported on the overall effectiveness of CHM [19,20,21,22,23,24,25,26,27,28,30,31,32,33,34,35,38,39,40,41,42,43,44,45,46,49,50,55,56,57,59,61,63,64,65,67,68]. Our meta-analysis demonstrated that the overall effectiveness was significantly higher in patients receiving CHM than for patients in control groups (RR 1.29, 95% CI 1.23 to 1.35) (Figure 5a). Relative to control groups, use of CHM significantly increased the overall effectiveness of UP treatments for both UP patients undergoing dialysis (RR 1.24, 95% CI 1.18 to 1.30) and patients not undergoing dialysis (RR 1.96, 95% CI 1.32 to 2.90) (Figure 5b). Moreover, compared to control groups, CHM demonstrated significant higher overall effectiveness in dialysis patients among all durations of treatment for <8 weeks of treatment (RR 1.29, 95% CI 1.20 to 1.38), ≥8 weeks of treatment (RR 1.23, 95% CI 1.10 to 1.37), and for ≥12 weeks of treatment (RR 1.28, 95% CI 1.19 to 1.38) (Figure 5c). Moreover, various Chinese herbal formulas significantly increased overall effectiveness of reducing symptoms of UP, including Touxie-Jiedu-Zhiyang decoction (RR 1.32, 95% CI 1.13 to 1.53), UCG (RR 1.28, 95% CI 1.18 to 1.38), and other decoctions (RR 1.26, 95% CI 1.19 to 1.33) (Figure 5d). Heterogeneity of scores was not significant for overall effectiveness, UP patients undergoing and not undergoing dialysis, duration of treatment, and effectiveness of all Chinese herbal formulas (overall effectiveness: I^2^ = 22%, *p* = 0.12; UP patients undergoing dialysis: I^2^ = 0%, *p* = 0.53; UP patients not undergoing dialysis: I^2^ = 27%, *p* = 0.26; <8 weeks of treatment: I^2^ = 0%, *p* = 0.65; ≥8 weeks of treatment: I^2^ = 44%, *p* = 0.09; ≥12 weeks of treatment: I^2^ = 0%, *p* = 0.84; UCG: I^2^ = 0%, *p* = 0.94; Touxie-Jiedu-Zhiyang decoction: I^2^ = 0%, *p* = 0.58; other decoctions: I^2^ = 18%, *p* = 0.22).

#### 2.3.3. Pittsburgh Sleep Quality Index (PSQI), Quality of Life (QOL)

We assessed sleep quality and quality of life with the PSQI and QOL scale in three [54,55,68] and four studies [31,49,61,68], respectively. The PSQI declined significantly in UP patients after CHM treatment (MD −2.20, 95% CI −2.77 to −1.64), and the score of the QOL scale increased significantly (MD 7.65, 95% CI 2.71 to 12.59) (Figure 6a,b). Heterogeneity was high for the QOL scale (I^2^ = 96%, *p* < 0.00001), whereas heterogeneity for PSQI was not significant. (I^2^ = 0%, *p* = 0.54).

### 2.4. Secondary Outcomes—Effects of Chinese Herbal Medicine on Laboratory Parameters

CHM significantly decreased the serum level of P (MD −0.20, 95% CI −0.28 to −0.13) and PTH (MD −76.68, 95% CI −115.62 to −37.74) (Appendix A). Regarding indicators of renal function, CHM was significantly related to lower serum concentrations of SCr (MD −52.31, 95% CI −93.32 to −11.31), and BUN (MD −1.97, 95% CI −3.67 to −0.26), but were significantly related to higher concentrations of eGFR (MD 2.82, 95% CI 0.65 to 4.99) (Appendix A). In assessing for inflammation, CHM was significantly related to decreased concentrations of CRP (MD −1.90, 95% CI −2.52 to −1.27), TNF-α (MD −16.88, 95% CI −19.35 to −14.41), β2-MG (MD −4.90, 95% CI −6.78 to −3.02), and IL-6 (MD −3.36, 95% CI −5.26 to −1.45) (Appendix A). Significantly elevated levels of hemoglobin (MD 4.52, 95% CI 0.23 to 8.80, I^2^ = 85%) were observed after administering CHM to UP patients (Appendix A).

Heterogeneities across trials were high for Ca (I^2^ = 94%, *p* < 0.00001), P (I^2^ = 86%, *p* < 0.00001), PTH (I^2^ = 98%, *p* < 0.00001), iPTH (I^2^ = 92%, *p* < 0.00001), SCr (I^2^ = 96%, *p* < 0.00001), BUN (I^2^ = 93%, *p* < 0.00001), UA (I^2^ = 98%, *p* < 0.00001), CRP (I^2^ = 89%, *p* < 0.00001, β2-MG (I^2^ = 99%, *p* < 0.00001), IL-6 (I^2^ = 96%, *p* < 0.00001), albumin (I^2^ = 86%, *p* < 0.00001), and hemoglobin (I^2^ = 85%, *p* < 0.00001), whereas heterogeneities for K (I^2^ = 0%, *p* = 0.34), eGFR (I^2^ = 0%, *p* = 0.87), TNF-α (I^2^ = 0%, *p* = 0.87), AST and ALT (AST: I^2^ = 43%, *p* = 0.15; ALT: I^2^ = 38%, *p* = 0.18) were not significant. However, changes in K, Ca, iPTH, UA, liver enzymes (ALT, AST), and albumin were not found to be significant between CHM administration and controls.

### 2.5. Adverse Drug Reactions

No significant increase in ADRSs was observed in patients after using CHM (RR 0.60, 95% CI 0.22 to 1.63) (Appendix A). Heterogeneity for ADRSs was high (I^2^ = 60%, *p* = 0.01).

### 2.6. Publication Bias

We conducted funnel plots to detect publication bias of VAS scores and overall effectiveness (Appendix A). Both funnel plots were asymmetrically distributed, demonstrating potential publication bias in our study.

### 2.7. Quality of Evidence

Given the high risk of bias for primary outcomes, the quality of evidence was low for assessing the efficacy of CHM in ameliorating symptoms associated with UP in patients (Appendix A).

## 3. Discussion

Our meta-analysis suggests that CHM significantly reduces various pruritus scores in UP patients (VAS, Duo, and Dirk R. Kuypers itch-intensity scores), improves sleep quality (PSQI) and quality of life (QOL), renal function (eGFR, BUN, and SCr), and alleviates inflammation (CRP, TNF-α, β2-MG, and IL-6). Different Chinese herbal formulas (Touxie-Jiedu-Zhiyang decoction, UCG, and other decoctions) were associated with significant reductions in the severity of pruritus in overall effectiveness and VAS scores. Compared to control groups, CHM significantly increased the overall effectiveness of relieving symptoms in UP patients both undergoing dialysis and not undergoing dialysis. In dialysis patients, CHM demonstrated significantly higher overall effectiveness for periods from less than 8 weeks to over 12 weeks. We expect that longer-duration treatments with CHM should further alleviate symptoms (decrease VAS scores) in patients. Our review detected no significant increase in ADRS after administering CHM to UP patients.

Based on the theory of Chinese medicine, UCG is used to improve intestinal motility, promote blood circulation, and remove pathogenic ingredients including toxins, dampness, and stasis [43]. Shi et al., showed that Touxie-Jiedu-Zhiyang decoction is used to remove toxins, invigorate qi, and replenish blood [50]; other decoctions, such as Si-Wu decoction and Zhi-Yang decoction are used to alleviate UP by nourishing blood and dispelling wind [57,60].

Regarding the unclear mechanism and pathogenesis of UP, a literature review proposed possible mechanisms of UP, including central stimulus from opioid receptors, deposited toxins, and systemic inflammation associated with histamines and proinflammatory cytokines such as CRP and IL-6 [14].

Xue et al. [69] reported that Chinese herbal bath therapy improves pruritus, decreases VAS scores, and increases effectiveness scores in UP patients. Moreover, the herbs most commonly used in bath therapies to treat UP patients are Difuzi, Baixianpi, Kushen, Chantui, Danggui, Xixin, Chuanxiong, Jingjie, Tufulin, and Dahuang [70], comprised of ingredients similar to the CHM reviewed in our study.

Our study suggests that the Touxie-Jiedu-Zhiyang decoction ameliorated UP symptoms. Huangqi, Dahuang, and Baishao are several important herbs comprising the Touxie-Jiedu-Zhiyang decoction [48]. Huangqi is beneficial for alleviating inflammation by decreasing TNF-α levels and suppressing the expression of Th2 cytokines in topical treatments [71]. Rhubarb (Dahuang), used as a laxative, is used for alleviating constipation [72] and showing nephroprotective effects in CKD [73]. Baishao has been shown to reduce inflammation by significantly inhibiting cAMP-phosphodiesterase (PDE) activity [74] and by displaying synergistic anti-inflammatory effects with Huangqin (another herb in the formula) in a cell model [75]. Huangqi, Dahuang, Fuling, and Danshen are important ingredients in UCG [42]. Fuling (Poria cocos) regulates by activating Th1 and alleviating Th2 immune response in murine tumor models [76]. Cryptotanshinone (CRT), extracted from Danshen (Salvia miltiorrhiza), was reported to possess anti-inflammation properties and alleviate pruritus by mitigating proinflammatory cytokines, such as TNF α and IL-1β, and by inhibiting mast cell degranulation [77]. Other decoctions contain common herbs, such as Danggui and Chantui, often used to alleviate skin disorders. Topical application of Danggui (Angelica sinensis) has been reported to attenuate inflammation and severity of pruritus symptoms by reducing the number of mast cells, serum IgE concentrations, and by reducing the concentration of inflammatory cytokines, such as IL-6, TNF-α, and IFN-γ [78]. Chantui (Cicadidae Periostracum) has been found to reduce IgE and histamine concentrations and suppress NLRP3, all of which are thought to help alleviate inflammation and itching sensations caused by UP [79]. Although the most beneficial combination of herbs, dosages, and routes for administering CHM should be further explored, we believe that CHM can be successfully used as a potential complementary treatment for UP symptoms.

Our meta-analysis revealed that CHM reduced symptoms of UP with a time-dependent tendency. Yang et al. [80] found that Gan-Lu-Yin, a Chinese herbal formula, significantly decreases the mRNA expression of TNF-α in a time-dependent manner. Paeonol, an extraction from CHM, has been shown to attenuate solar UV-induced skin inflammations by decreasing T-LAK cell-originated protein kinase (TOPK) activity in a time-dependent manner [81]. A variety of CHM ingredients display anti-inflammation effects over time, which might explain why longer durations of CHM treatments are associated with improved alleviation of symptoms in UP in our included studies.

In our study, CHM improved quality of life and sleep quality of UP patients. Several studies suggested that UP might reduce quality of life and increase sleep disturbances in patients [82,83,84]. Traditional Chinese medicine (TCM), such as acupuncture, has been shown to improve hemodialysis-related complications, including symptoms of UP, and insomnia in chronic kidney disease (CKD) patients, by regulating the sympathetic nervous system [85]. Cochrane’s systematic review [86] demonstrated that manual acupressure significantly reduced depression and improved sleep quality and fatigue in patients with CKD, although those results had very low quality of evidence. Based on the previous reports, we suggest that TCM, including CHM, acupressure, and acupuncture is beneficial for UP patients, whereas a higher quality of studies should be conducted to verify the evidence.

CHM appears to play a role in balancing concentrations of Ca and P, improving renal functions, and delaying progression in chronic renal failure in patients [87,88]. PTH has also been associated with mast cell activation, which releases histamine and causes pruritus [63]. A rat model demonstrated that CHM combined with acupoint thread implantation could reduce PTH concentrations in rats with CKD [89]. In our meta-analysis, CHM improved renal function and efficacy in UP patients both undergoing and not undergoing dialysis. Wang et al. [90] found that traditional Chinese medicines improved eGFR and hemoglobin in stage III CKD patients. In two meta-analyses, UCG were also shown to significantly reduce SCr and increase eGFR in stages III-V CKD [91] and in dialysis patients [14]. Huangqi (Astragalus membranaceus), an important herb in Touxie-Jiedu-Zhiyang decoctions and UCG, has been reported to reduce proteinuria and SCr, while also increasing albumin and hemoglobin in CKD patients [92]. In our meta-analysis, CHM appeared to improve renal function in UP patients by significantly lowering SCr, BUN, P, and PTH levels and increasing eGFR. However, no significant difference after CHM administration was noted in Ca concentrations, a finding that is similar to the meta-analysis conducted by Lu et al. [14]. More studies should be conducted to clarify how CHM ameliorates symptoms in CKD patients and compare different responses in patients undergoing and not undergoing dialysis.

Our meta-analysis suggested that CHM significantly reduces inflammation in patients with UP. Xuebijing injections, which are composed of five Chinese herbal extracts (including Honghua, Chishao, Danggui, Chuanxiong, and Danshen), are similar to ingredients in Touxie-Jiedu-Zhiyang decoctions and in UCG. Xuebijing injections attenuated renal inflammation and reduced levels of IL-6 and TNF-α in a mice model [93]. One meta-analysis demonstrated that an injection of ligustrazine (a compound extracted from Chuanxiong) and Danshen (Salvia miltiorrhiza) appeared to reduce inflammation in diabetic kidney diseases [94]. In addition, previous reviews also demonstrated that elevated IL-31 is associated with UP in dialysis patients [95,96]. Furthermore, Wang et al., demonstrated that a Chinese herbal formula, Yangxue-Runfu-Yin, significantly lowers the level of IL-31, ameliorates pruritus severity, and improves sleep and quality of life in hemodialysis patients. All these findings suggest that CHM could ameliorate UP symptoms by improving renal function and by attenuating inflammation.

Hypoalbuminemia is common in patients on dialysis, which is associated with malnutrition and inflammation [97,98]. Huangqi, an important herb in Touxie-Jiedu-Zhiyang decoction and UCG, has been shown to be beneficial in alleviating nephrotic syndrome by increasing plasma albumin and reducing excretion of urine albumin [99]. However, one study showed no significant difference in serum albumin levels between UP and non-UP patients undergoing dialysis [100]. Besides hypoalbuminemia, CKD patients often develop renal anemia [101]. One study reported therapeutic effects on renal fibrosis and renal anemia after providing UCG, which was likely achieved by modulating transforming growth factor-β and erythropoietin signaling pathways in a mouse model [102]. Yin et al., [90] found that Niaoduqing granules increased hemoglobin level [91]. Consistent with results of previous meta-analyses, our study suggests that the administration of CHM also helps increase hemoglobin and albumin concentrations.

ADRSs are commonly reported in the treatment of UP with CHM; the ADRSs include nausea, vomiting, allergy, headache, and dizziness. Mild diarrhea, nausea, and abdominal discomfort have also been identified following treatment with the Shufeng-Liangxue decoction, a drug similar in formulation to the Touxie-Jiedu-Zhiyang decoction [103]. Regarding herb-induced liver injury, a recent systematic review [104] revealed He-Shou-Wu has been reported as a culprit of herb-induced liver injuries. However, we found no significant elevation of liver enzymes in our meta-analysis. Additional studies should examine possible ADRSs in the treatment of CHM in UP patients.

Selected studies in our meta-analysis showed heterogeneity in response to certain clinical factors, as outlined below. First, heterogeneity in efficacy was associated with different types of Chinese herbal formulas, such as in other decoction groups. Second, frequencies and dosages of CHM administered differed across the studies. Third, there were some discrepancies in the interventions of the control groups. Lastly, our meta-analysis included studies using different tools for pruritus assessment, which may lead to heterogeneity.

There were several limitations to our study. First, the method of randomization applied in most of the studies was unclear. No trial reported double-blinding. Second, the included RCTs were a small sample size. Third, only a few studies could be included when performing subgroup analyses due to the different components of various Chinese herbal formulas. Fourth, inconsistent symptomatic treatments and lack of appropriate controls might lead to a modest reduction of VAS score in the CHM. However, higher quality studies should be executed that involve examining a larger population of data sets and examining the efficacy of head-to-head comparisons among different Chinese herbal formulas.

## 4. Materials and Methods

We searched seven databases from their inception to 3 May 2022: PubMed, Embase, Cochrane Library, CINAHL, Chinese National Knowledge Infrastructure, Airiti library, and Wanfang. We used MeSH and Emtree search headings, as follows: Chinese medicine (including herbal medicine, pill, powder, san, granule, and formula), pruritus, uremia, chronic kidney disease, dialysis, and their synonyms. We searched for free text words using these terms and their combinations (Appendix A). In addition, we manually searched the reference sections of accessed papers and contacted known experts in the field to identify other studies. Finally, unpublished studies were inspected from the ClinicalTrials.gov registry (http://clinicaltrials.gov/, accessed on 1 August 2022). Our search was not restricted by language, and our method of systematic review was deemed acceptable by the online PROSPERO registry of the National Institute for Health Research (CRD 42022334701).

RCTs were included to evaluate the efficacy of CHM for UP patients. Our predetermined inclusion criteria included patients with UP, administration of oral CHM to patients, and the availability of quantitative data to assess pruritus severity. We excluded review articles, studies examining other traditional Chinese medicine interventions (e.g., acupuncture, acupressure, herbal bath, enema), and studies of patients not diagnosed with UP. We included studies in our analysis without regard to the type of pruritus evaluations utilized. To obtain raw or missing data in specific studies, we contacted investigators of those studies by e-mail.

Two reviewers (Chien-Cheng Lai and Ping-Hsun Lu) independently extracted the following information from each study: first author, publication year, sample size, age, period of intervention, dosage and frequency of interventions and comparisons thereof, specific means for assessing pruritus severity, quantified data on pruritus severity, quality of life and sleep quality indices, rates of overall effectiveness, and rates of adverse events. Other laboratory data were also extracted, including renal function, inflammation biomarkers, and serum concentrations for electrolytes and hormones.

The preliminarily selected studies were assessed for eligibility for meta-analysis by the two reviewers according to the above-listed inclusion criteria. The decisions of the two reviewers were individually recorded and compared, and any disagreement was resolved by a third reviewer (Po-Hsuan Lu). The risk of bias for the selected RCTs was evaluated with Cochrane Collaboration’s Risk of Bias 2 tool [105].

We evaluated the efficacy of CHM using outcome measures as described below. The primary outcomes examined included mean difference (MD) in: VAS scores, Duo pruritus scores, scores on the Dirk R. Kuypers itching scale, quality of life and sleep quality indices, and the risk ratios (RR) for overall effectiveness. Secondary outcomes included the mean differences in serum concentrations of albumin, hemoglobin, electrolytes (K, Ca, and P), enzymes, and hormones (ALT, AST, and PTH), renal function index (SCr, eGFR, and BUN), inflammation biomarkers (CRP, TNF-α, and B2-MG, IL-6), and the RR of adverse event rates. We measured dichotomous outcomes as RR and continuous outcomes as weighted mean differences (WMDs). Both summary statistics were reported with 95% CIs. We conducted our meta-analysis using the RevMan 5.4 software (Cochrane Collaboration, Copenhagen, Denmark). Our meta-analysis was conducted following recommendations of the Preferred Reporting Items for Systematic Reviews and Meta-Analyses (PRISMA) guidelines [106]. The I^2^ statistic and Cochran Q statistic were used to quantify statistical heterogeneity across the included studies, whereby substantial heterogeneity was detected when the I^2^ statistic was > 50% or probability (*p*) was < 0.1. Considering clinical heterogeneity, we performed a random-effects model meta-analysis. Subgroup analyses were performed to assess between-group differences and explain the heterogeneity. We conducted funnel plots to detect publication bias. Certainty of evidence was assessed using the Grading of Recommendations Assessment, Development, and Evaluation (GRADE) approach [107].

## 5. Conclusions

This systematic review and meta-analysis demonstrates that CHM, including Touxie-Jiedu-Zhiyang decoctions, UCG, and other decoctions reduce pruritus severity based on overall effectiveness and VAS scores. In addition, adjunctive CHM improves quality of life, renal function, and attenuates inflammation, whereas no statistically significant difference in adverse drug reaction is found compared to UP patients only who received hemodialysis alone or with antipruitic treatments. Compared to control groups, CHM increases overall effectiveness in both UP patients undergoing dialysis and those not undergoing dialysis. In dialysis patients, CHM alleviates UP and reduces the VAS score over time, especially after more than 12 weeks of use. However, for future research, we recommend examining studies with more patients and higher-quality studies that focus on head-to-head comparisons among CHM interventions in UP patients.

## Figures and Tables

**Figure 1 pharmaceuticals-15-01239-f001:**
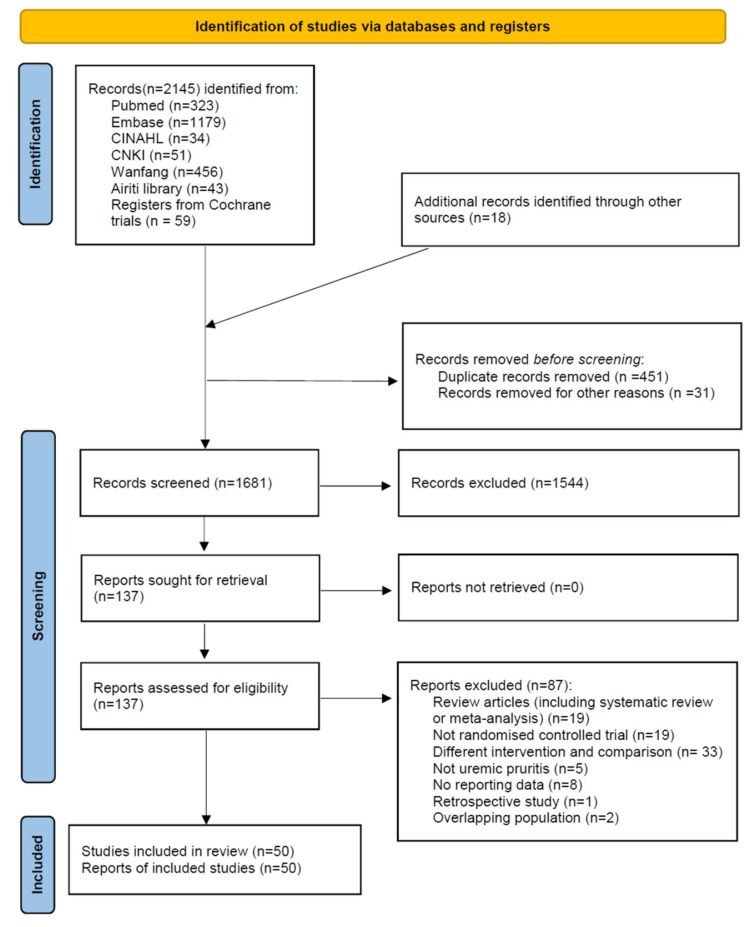
PRISMA 2020 flow diagram.

**Figure 2 pharmaceuticals-15-01239-f002:**
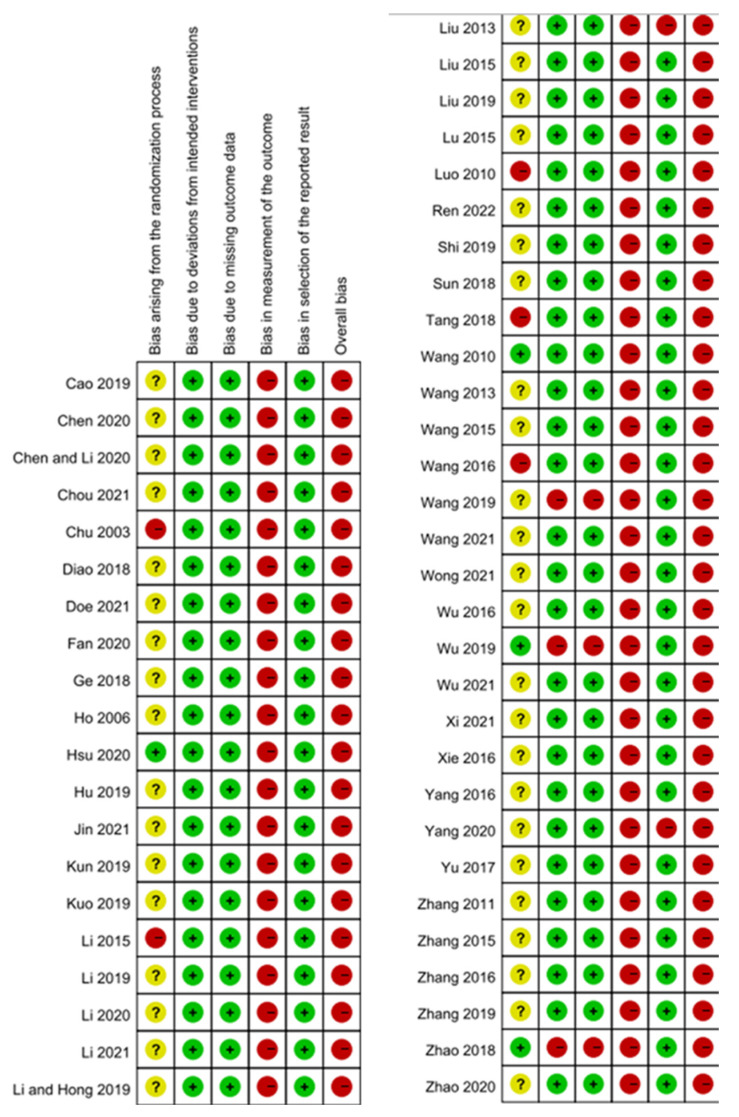
Risk of bias summary (“+” = low risk of bias, “− “ = high risk of bias, “?” = unclear risk of bias).

**Figure 3 pharmaceuticals-15-01239-f003:**
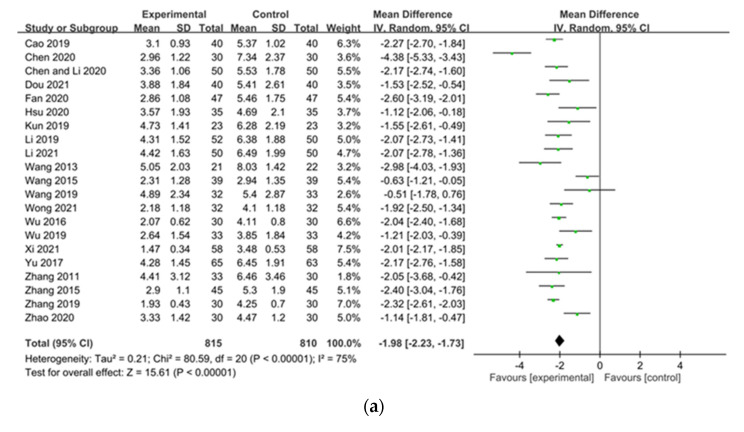
Forest plot of pruritus score (visual analog scale (VAS)) in patients with uremic pruritus treated with Chinese herbal medicine (CHM): (**a**) VAS score in included studies; (**b**) VAS score in durations of CHM treatment in dialysis patients; (**c**) VAS score in different Chinese herbal formulas.

**Figure 4 pharmaceuticals-15-01239-f004:**
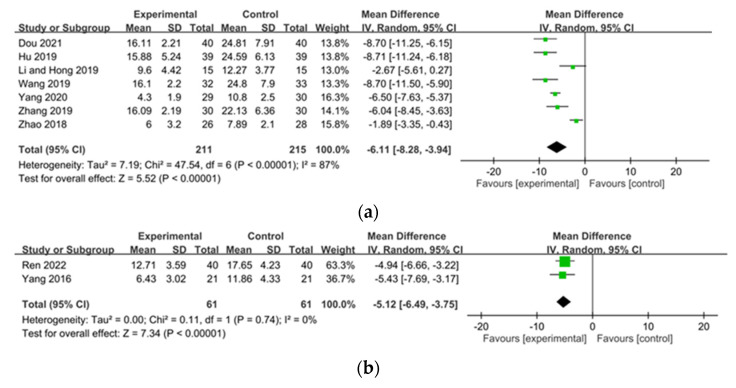
Forest plot of pruritus score: (**a**) Duo; (**b**) Dirk R. Kuypers itching scale in patients with uremic pruritus treated with Chinese herbal medicine.

**Figure 5 pharmaceuticals-15-01239-f005:**
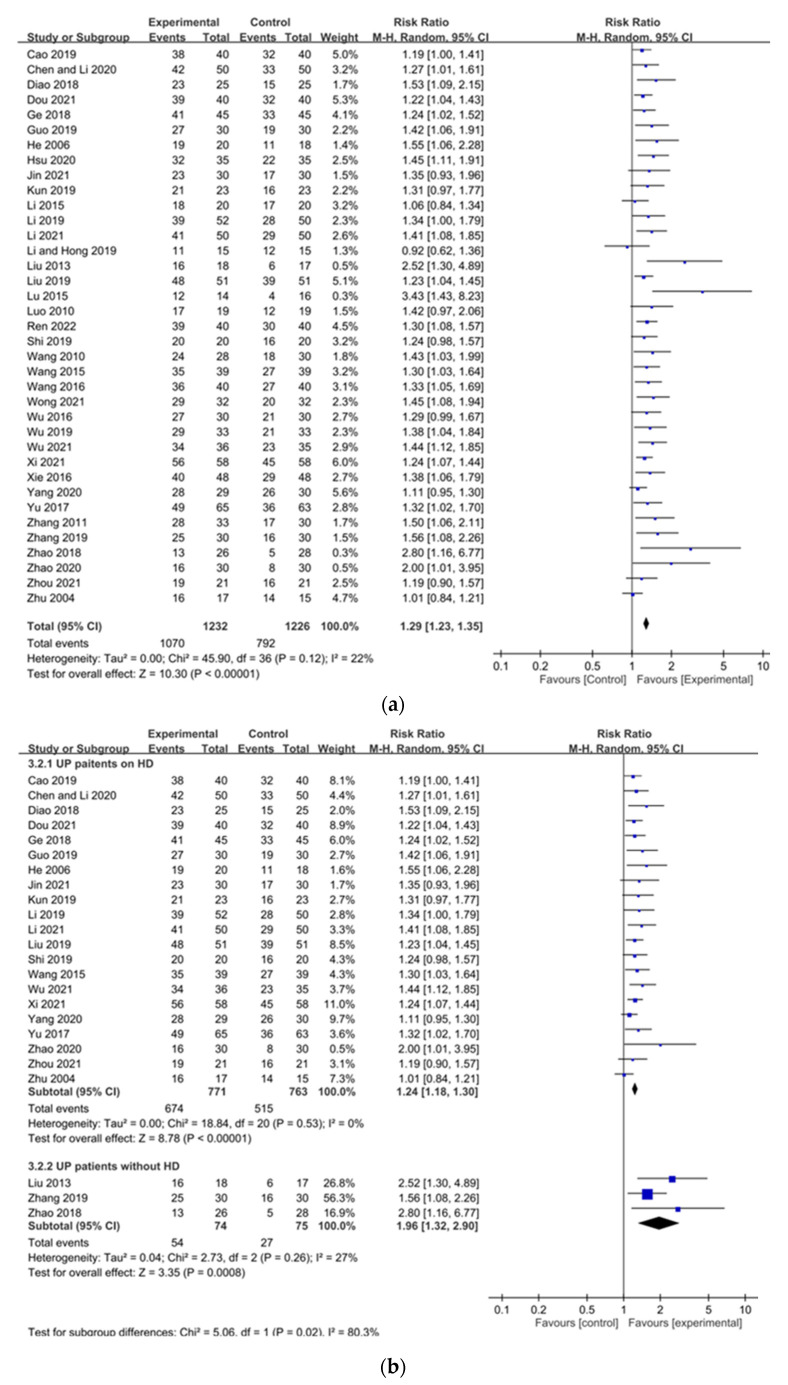
Forest plot of overall effectiveness of patients with uremic pruritus treated with Chinese herbal medicine (CHM): (**a**) overall effectiveness of included studies; (**b**) overall effectiveness of patients with uremic pruritus undergoing dialysis or not undergoing dialysis; (**c**) overall effectiveness of durations of CHM treatment in dialysis patients; (**d**) overall effectiveness of different Chinese herbal formulas.

**Figure 6 pharmaceuticals-15-01239-f006:**
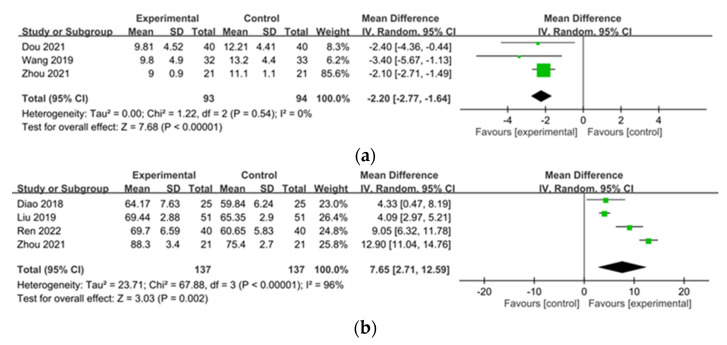
Forest plot of quality of life: (**a**) Pittsburgh Sleep Quality Index (PSQI); (**b**) Quality of Life (QOL) scale in patients with uremic pruritus treated with Chinese herbal medicine.

**Table 1 pharmaceuticals-15-01239-t001:** Characteristics of selected studies.

Study (Year)	Control/ Comparison ^a^	Intervention/ Exposure ^a^	No. of Patients (I/C)	Age (Years)	Dosage and Frequency	Duration	Pruritus Severity Assessment	Pruritus Score (Before → After)
**Uremic clearance granule (UCG)**								
Yang (2016) [36]	HD/HD + HP	UCG + HD/HD + HP	21/21	I: 51.48 (13.49) C: 51.67 (11.68)	2.5 g, 2 times/d	1 M	Kuypers PS	I: 11.57 (2.45) → 6.43 (3.02) C:11.67 (4.98) → 11.86 (4.33)
Sun et al. (2018) [37]	HD	UCG + HD	54/54	I: 54.12 (5.78) C: 54.08 (6.23)	2.5 g, 2 times/d	NA	NA	I: 11.56 (3.02) → 5.12 (0.89) C:11.89 (3.12) → 8.28 (2.02)
Guo et al. (2019) [38]	HD	UCG + HD	30/30	I: 42.6 (3.2) C: 41.9 (3.4)	5 g, 4 times/d	3 M	NA	NA
Yu et al. (2017) [39]	High-flux HD	UCG + High-flux HD	65/63	I: 35-68 C: 36–70	5 g, 4 times/d	3 M	VAS	I: 8.17 (1.94) → 4.28 (1.45) C: 8.21 (1.78) → 6.45 (1.91)
Cao (2019) [40]	High-flux HD	UCG + High-flux HD	40/40	I: 59.7 C: 59.8	5 g, 4 times/d	2 M	NRS	I: 7.89 (1.31) → 3.10 (0.93) C: 7.95 (1.43) → 5.37 (1.02)
Kun et al. (2019) [41]	High-flux HD	UCG + High-flux HD	23/23	I: 45.3 (5.3) C: 46.1 (4.9)	5 g, 4 times/d	3 M	VAS	I: 8.13 (1.77) → 4.73 (1.41) C: 8.40 (2.07) → 6.28 (2.19)
Li et al. (2019) [42]	High-flux HD	UCG + High-flux HD	52/50	I: 47.2 (3.7) C: 46.7 (4.2)	5 g, 4 times/d	3 M	VAS	I: 8.18 (1.69) → 4.31 (1.52) C: 8.20 (1.96) → 6.38 (1.88)
Chen and Li et al. (2020) [43]	High-flux HD	UCG + High-flux HD	50/50	I: 65.72 (10.33) C: 64.12 (10.54)	5 g, 4 times/d	3 M	VAS	I: 7.62 (1.02) → 3.36 (1.06) C: 7.54 (0.98) → 5.53 (1.78)
							5-D Itch Scale	I: 17.37 (3.56) → 6.44 (1.59) C: 16.98(3.72) → 10.82 (2.31)
							DLQI	I: 21.84 (5.53) → 8.36 (2.21) C: 21.54 (5.70)→10.55 (3.88)
Xi (2021) [44]	High-flux HD	UCG + High-flux HD	58/58	I: 47.88 (3.52) C: 47.79 (3.41)	5 g, 4 times/d	NA	VAS	I: 7.21 (1.72) → 1.47 (0.34) C: 7.23 (1.71) → 3.48 (0.53)
Li (2021) [45]	High-flux HD	UCG + High-flux HD	50/50	I: 51.21 (1.92) C: 49.39 (2.74)	5 g, 4 times/d	14 Weeks	NA	I: 8.29 (1.70) → 4.42 (1.63) C: 8.31 (2.07) → 6.49 (1.99)
**Touxie-Jiedu-Zhiyang Decoction**								
Wang et al. (2015) [46]	HD	Touxie-Jiedu-Zhiyang Formula + HD	39/39	I: 49 (8) C: 52 (10)	100 mL, 2 times/d	3 M	VAS	I: 6.95 (1.47) → 2.31 (1.28) C: 6.87 (1.53) → 2.94 (1.35)
Zhang et al. (2015) [47]	Antihistamine + Emulsifying oil +HD	Touxie Zhiyang Decoction + HD	45/45	NA	NA	NA	VAS	I: 7.2 (2.1) → 2.9 (1.1) C: 7.3 (2.0) → 5.3 (1.9)
Zhang et al. (2016) [48]	HD	Touxie-Jiedu-Zhiyang Decoction + HD	50/50	I: 58.43 (12.82) C: 59.84 (13.76)	NA, 2 times/d	3 M	VAS	NA
Diao et al. (2018) [49]	HD	Touxie-Jiedu-Zhiyang Decoction + HD	25/25	I: 62.3 (4.8) C: 61.6 (5.4)	NA, 2 times/d	3 M	TCM new drug clinical research guideline	I: 2.51 (0.79) → 0.72 (0.34) C: 2.47 (0.82) → 1.88 (0.45)
Shi (2019) [50]	HD	Touxie-Jiedu-Zhiyang Decoction + HD	20/20	I: 45.24 (2.78) C: 45.21 (2.42)	NA, 2 times/d	3 M	NA	NA
Chen (2020) [51]	HD	Touxie-Jiedu-Zhiyang Decoction + HD	30/30	I: 56.13 (7.45) C: 56.34 (7.12)	NA, 2 times/d	3 M	VAS	I: 7.35 (2.13) → 2.96 (1.22) C: 7.32 (2.24) → 7.34 (2.37)
**Yangxue-Runfu-Yin**								
Liu (2015) [52]	Loratadine + HD	Yangxue-Runfu-Yin + Loratadine + HD	20/20	I: 57.65 (3.21) C: 56.81 (3.04)	NA	NA	NA	NA
Hu (2019) [53]	Loratadine + HD	Modified Yangxue-Runfu-Yin + HD	39/39	I: 61.05 (7.45) C: 60.86 (7.32)	1 pack, 1 times/d	NA	Duo PS	I: 32.82 (4.33) → 15.88 (5.24) C: 32.15 (3.46) → 24.59 (6.13)
Wang et al. (2019) [54]	Loratadine + HD	Modified Yangxue-Runfu-Yin + HD	32/33	I: 50.7 (16.9) C: 52.2 (10.8)	1 pack, 2 times/d	2 M	VAS	I: 8.01(2.25) → 4.89(2.34) C: 7.73(1.53) → 5.40(2.87)
							Duo PS	I: 31.20 (8.90) → 16.10 (2.20) C: 28.90 (9.20) → 24.80 (7.90)
Dou (2021) [55]	Desloratadine + HD	Modified Yangxue-Runfu-Yin + Desloratadine + HD	40/40	I: 53.56 (15.67) C: 53.62 (15.48)	100 mL, 2 times/d	0.5 M	VAS	I: 8.02 (2.26) → 3.88 (1.84) C: 7.74 (1.54) → 5.41 (2.61)
							Duo PS	I: 31.21 (8.91) → 16.11 (2.21) C: 28.91 (9.21) → 24.81 (7.91)
**Other Chinese herbal decoctions**								
Zhu et al. (2004) [56]	Vit A. + Topical tincture + Cetirizine + HD	Yangxue Wensheng Decoction + Vit A. + Topical tincture + Cetirizine + HD	17/15	I: 70.2 (4.3) C: 71.6 (3.1)	NA, 2 times/d	2 M	NA	NA
He (2006) [57]	Cetirizine + HD	Siwu Tang + Erzhi Wan + Cetirizine + HD	20/18	I: 50.5 C: 49.8	NA, 2 times/d	1 Week	NA	NA
Wang et al. (2013) [58]	HD/HD + HP	Xiaoyang Particles + HD	21/22	NA	1 pack, 2 times/d	2 M	VAS	I: 8.08 (1.02) → 5.05 (2.03) C: 7.89 (1.32) → 8.03 (1.42)
Ge (2018) [59]	Loratadine + HD	Xiaofeng Zhiyang Particles + Loratadine + HD	45/45	I: 56.03 (7.26) C: 55.24 (7.31)	3 packs, 2 times/d	0.5 M	TCM new drug clinical research guideline	I: 102.37 (16.87) → 40.32 (20.16) C: 99.26 (17.45) → 64.21 (25.02)
Tang et al. (2018) [60]	Gabapentin + HD	Zhiyang Decoction + Gabapentin + HD	18/18	I: 57.33 (16.45) C: 58.5 (16.21)	NA, 2 times/d	1 M	TCM new drug clinical research guideline	NA
Liu et al. (2019) [61]	Cetirizine + HD	Jingfu Zhiyang Particles + Cetirizine + HD	51/51	I: 55.43 (11.02) C: 55.47 (11.01)	6 g, 3 times/d	1 M	Self-made PS questionnaire	I: 16.26 (4.49) → 6.01 (3.54) C: 16.33 (4.51) → 9.73 (3.55)
Fan (2020) [62]	Emulsifying oil + HD	Touxie-Jiedu-Zhiyang Decoction + Emulsifying oil + HD	47/47	I: 27.32 (2.13) C: 27.37 (2.42)	500 mL, 2 times/d	NA	VAS	I: 7.31 (2.11) → 2.86 (1.08) C: 7.17 (2.16) → 5.46 (1.75)
Yang et al. (2020) [63]	HD + HP	Baifuzhi Weiliang Decoction + HD + HP	29/30	I: 49.1 (8.5) C: 49.5 (8.2)	150 mL, 2 times/d	3 M	Sergio PS	I: 30.9 (8.8) → 4.3 (1.9) C: 30.4 (8.6) → 10.8 (2.5)
Zhao (2020) [64]	HD	Siwu Decoction + HD	30/30	I: 61.17 (13.35) C: 58.83 (14.61)	NA, 2 times/d	1 M	VAS	I: 6.23 (1.22) → 3.33 (1.42) C: 6.23 (1.63) → 4.47 (1.20)
Jin et al. (2021) [65]	High-flux HD	Mahuang Lianqiao Chixiaodou + Yiyifuzhi baijiang Decoction + High-flux HD	30/30	I: 53.26 (11.38) C: 53.26 (11.38)	100 mL, 2 times/d	6 M	Sergio PS	NA
Wang et al. (2021) [66]	HD + HP	Feng Xueqing Yin +HD + HP	16/16	I: 57.19 (5.79) C: 53.50 (9.14)	1 pack,2 times/d	1 M	VAS	I: 39.21 (2.50) → 17.08 (3.05) C: 39.91 (2.76) → 29.06 (2.86)
Wu et al. (2021) [67]	HD	Modified Qufeng Decoction + HD	36/35	I: 45.8 (8.4) C: 46.3 (8.6)	100 mL, 2 times/d	1 M	VAS	I: 27.65 (3.24) → 4.18 (1.20) C: 27.49 (3.20) → 12.84 (3.62)
Zhou et al. (2021) [68]	HD + HP	Chinese herbal medicine ^b^ + HD + HP	21/21	I: 43.59 (3.72) C: 43.46 (3.68)	150 mL, 2 times/d	2M	NA	NA
**Other Chinese herbal decoctions—Control group with additional treatment**								
Wang et al. (2010) [19]	Chlorphenamine + HD	Jiebiao Qufengzhiyang particles + HD	28/30	I: 47 (11) C: 46 (9)	5 g, 2 times/d	0.5 M	NA	NA
Luo et al. (2010) [20]	Calamine lotion + HD	Chinese herbal medicine ^c^ + HD	19/19	NA	100 mL, 2 times/d	3 M	NA	NA
Zhang et al. (2011) [21]	Loratadine + HD	Qingxin Lianxi In + HD	33/30	I: 59.7 (12.4) C: 60.9 (11.7)	50 mL, 2 times/d	20 Days	VAS	I: 8.75 (2.61) → 4.41 (3.12) C: 8.59 (2.86) → 6.46 (3.46)
Li (2015) [22]	Loratadine + HD	Modified Siwu Decoction + HD	20/20	I: 49.2 (1.2) C: 50.1 (1.5)	NA, 3 times/d	2 M	VAS	NA
Wang et al. (2016) [23]	Charcoal Tablets + HD	Shengyang Xiehuo Decoction + HD	40/40	NA	100 mL, 3 times/d	1 M	Criteria of diagnosis and therapeutic effect of TCM diseases and syndromes	NA
Wu (2016) [24]	Cetirizine + HD	Modified Jiedu Huoxue Decoction + HD	30/30	I: 45.8 (5.1) C: 44.6 (5.5)	100 mL, 3 times/d	1 M	VAS	I: 6.69 (1.57) → 2.07 (0.62) C: 6.75 (1.53) → 4.11 (0.80)
Xie (2016) [25]	HD + HP	Modified Siwu Decoction + HD	48/48	I: 43.26 (8.37) C: 44.15 (9.25)	100 mL, 3 times/d	2 M	Criteria of diagnosis and therapeutic effect of TCM diseases and syndromes	I: 25.63 (4.55) → 9.06 (4.32) C: 25.87 (5.06) → 17.63 (4.35)
Li and Hong et al. (2019) [26]	HD + HP	Danggui Yinzi + HD	15/15	I: 50.62 (28.52) C: 47.22 (20.13)	100 mL, 2 times/d	1 M	Sergio PS	I: 27.07 (8.17) → 9.60 (4.42) C: 25.60 (7.68) → 12.27 (3.77)
Wu (2019) [27]	Loratadine + HD	Wushe Rongpi Decoction + HD	33/33	I: 70 C: 71	50 mL, 2 times/d	2 M	VAS	I: 6.33 (1.81) → 2.64 (1.54) C: 6.30 (1.83) → 3.85 (1.84)
Hsu (2020) [28]	Loratadine + HD	Modified Danggui Yinzi + HD	35/35	I: 54.83 (11.40) C: 58.43 (12.08)	100 mL, 2 times/d	2 M	VAS	I: 6.17 (1.98) → 3.57 (1.93) C: 6.37 (2.20) → 4.69 (2.10)
Li et al. (2020) [29]	Calamine lotion + HD	Mahuang Lianqiao Chixiaodou decoction + HD	31/31	I: 59.3 (8.6) C: 59.6 (8.9)	NA	0.5 M	Li’s pruritus scale	I: 4.0 (0.9) → 1.9 (0.3) C: 4.1 (0.8) → 2.8 (0.4)
Wong (2021) [30]	Loratadine + HD	Taohong Danggui Yinzi + HD	32/32	I: 61.82 (11.58) C: 63.45 (11.99)	50 mL, 2 times/d	2 M	VAS	I: 6.46 (1.57) → 2.18 (1.18) C: 6.58 (1.73) → 4.10 (1.18)
Ren (2022) [31]	Loratadine + HD	Xiaofeng Zhiyang granules + HD	40/40	I: 52.32 (11.19) C: 52.37 (11.26)	18 g, 2 times/d	1 M	Kuypers PS	I: 22.17 (4.66) → 12.71 (3.59) C: 22.22 (4.88) → 17.65 (4.23)
**Patients with uremic pruritus without dialysis**								
Liu (2013) [32]	CT	Buyanghuanwu Tang + CT	18/17	I: 42.22 (9.64) C: 40.59 (9.51)	100 mL, 3 times/d	1 M	Modified Duo PS	NA
Lu (2015) [33]	CT	Modified JieDu ZhiYang Decoction + CT	16/14	I: 55.50 (11.38) C: 49.75 (14.89)	NA	1 M	NA	I: 4.43 (1.16) → 2.71 (0.99) C: 4.25 (1.24) → 4.13 (1.36)
Zhao (2018) [34]	CT	Qingjiangxiezhuo decoction + CT	26/28	I: 51.62 (9.64) C: 49.75 (7.73)	200 mL, 2 times/d	2 M	Modified Duo PS	I: 8.38 (1.86) → 6.00 (3.20) C: 8.57 (1.64) → 7.89 (2.10)
Zhang (2019) [35]	CT + TCM Patent Prescription	Yishen Huoxue Decoction + CT + TCM Patent Prescription	30/30	I: 54.37 (12.66) C: 55.53 (12.01)	75 mL, 2 times/d	2 M	VAS	I: 6.79 (1.49) → 1.93 (0.43) C: 6.45 (1.32) → 4.25 (0.70)
**Study (Year)**		**Overall Effectiveness**		**Pittsburgh Sleep Quality Index (PSQI) (Before → After)**			**Quality of Life (QOL) (Before → After)**	
**Uremic clearance granule (UCG)**								
Yang (2016) [36]		NA		NA			NA	
Sun et al. (2018) [37]		NA		NA			NA	
Guo et al. (2019) [38]		I: 27/30 C: 19/30		NA			NA	
Yu et al. (2017) [39]		I: 49/65 C: 36/63		NA			NA	
Cao (2019) [40]		I: 38/40 C: 32/40		NA			NA	
Kun et al. (2019) [41]		I: 21/23 C: 16/23		NA			NA	
Li et al. (2019) [42]		I: 39/52 C: 28/50		NA			NA	
Chen and Li et al. (2020) [43]		I: 42/50 C: 33/50		NA			NA	
Xi (2021) [44]		I: 56/58 C: 45/58		NA			NA	
Li (2021) [45]		I: 41/50 C: 29/50		NA			NA	
**Touxie-Jiedu-Zhiyang Decoction**								
Wang et al. (2015) [46]		I: 35/39 C: 27/39		NA			NA	
Zhang et al. (2015) [47]		NA		NA			NA	
Zhang et al. (2016) [48]		NA		NA			NA	
Diao et al. (2018) [49]		I: 23/25 C: 15/25		NA			I: 34.85 (11.92) → 64.17 (7.63) C: 35.04 (12.65) → 59.84 (6.24)	
Shi (2019) [50]		I: 20/20 C: 16/20		NA			NA	
Chen (2020) [51]		NA		NA			NA	
**Yangxue-Runfu-Yin**								
Liu (2015) [52]		NA		NA			NA	
Hu (2019) [53]		NA		NA			NA	
Wang et al. (2019) [54]		NA		I: 14.70 (6.10) → 9.80 (4.90) C: 14.50 (5.30) → 13.20 (4.40)			NA	
Dou (2021) [55]		I: 39/40 C: 32/40		I: 14.71 (6.11) → 9.81 (4.52) C: 14.51 (5.31) → 12.21 (4.41)			NA	
**Other Chinese herbal decoctions**								
Zhu et al. (2004) [56]		I: 16/17 C: 14/15		NA			NA	
He (2006) [57]		I: 19/20 C: 11/18		NA			NA	
Wang et al. (2013) [58]		NA		NA			NA	
Ge (2018) [59]		I: 41/45 C: 33/45		NA			NA	
Tang et al. (2018) [60]		NA		NA			NA	
Liu et al. (2019) [61]		I: 48/51 C: 39/51		NA			I: 55.85 (2.71) → 69.44 (2.88) C: 56.01 (3.76) → 65.35 (2.90)	
Fan (2020) [62]		NA		NA			I: 21.54 (2.34) → 52.16 (2.47) C: 21.43 (2.16) → 41.38 (2.43)	
Yang et al. (2020) [63]		I: 28/29 C: 26/30		NA			NA	
Zhao (2020) [64]		I: 16/30 C: 8/30		NA			NA	
Jin et al. (2021) [65]		I: 23/30 C: 17/30		NA			NA	
Wang et al. (2021) [66]		NA		NA			NA	
Wu et al. (2021) [67]		I: 34/36 C: 23/35		NA			NA	
Zhou et al. (2021) [68]		I: 19/21 C: 16/21		I: 15.5 (1.2) → 9.0 (0.9) C: 15.4 (1.2) → 11.1 (1.1)			I: 62.4 (3.3) → 88.3 (3.4) C: 62.4 (3.4) → 75.4 (2.7)	
**Other Chinese herbal decoctions—Control group with additional treatment**								
Wang et al. (2010) [19]		I: 24/28 C: 18/30		NA			NA	
Luo et al. (2010) [20]		I: 17/19 C: 12/19		NA			NA	
Zhang et al. (2011) [21]		I: 28/33 C: 17/30		NA			NA	
Li (2015) [22]		I: 18/20 C: 17/20		NA			NA	
Wang et al. (2016) [23]		I: 36/40 C: 27/40		NA			NA	
Wu (2016) [24]		I: 27/30 C: 21/30		NA			NA	
Xie (2016) [25]		I: 40/48 C: 29/48		NA			NA	
Li and Hong et al. (2019) [26]		I: 11/15 C: 12/15		NA			NA	
Wu (2019) [27]		I: 29/33 C: 21/33		NA			NA	
Hsu (2020) [28]		I: 32/35 C: 22/35		NA			NA	
Li et al. (2020) [29]		NA		NA			NA	
Wong (2021) [30]		I: 29/32 C: 20/32		NA			NA	
Ren (2022) [31]		I: 39/40 C: 30/40		NA			I: 52.80 (7.35) → 69.70 (6.59) C: 52.82 (7.28) → 60.65 (5.83)	
**Patients with uremic pruritus without dialysis**								
Liu (2013) [32]		I: 16/18 C: 6/17		NA			NA	
Lu (2015) [33]		I: 12/14 C: 4/16		NA			NA	
Zhao (2018) [34]		I: 13/26 C: 5/28		NA			NA	
Zhang (2019) [35]		I: 25/30 C: 16/30		NA			NA	

CT, conventional treatment; C, control group; DLQI, dermatology life quality index; HD, hemodialysis; HP, hemoperfusion; I, intervention group; M, month; NA, not applicable; NRS, numeric rating scale; *PS*, pruritus score; TCM, traditional Chinese medicine; UCG, uremic clearance granule; VAS, visual analog scale. ^a^ Conventional treatment (acid–base status with electrolyte balanced, sodium and fluid restriction, blood pressure maintenance) for chronic kidney disease in both intervention and control groups. ^b^ Chinese medicine, including Huangqi, Danggui, Danshen, Baishao, Baizhu, Difuzi, Baixianpi, Chuanxiong, Tufuling, Jingjie, Fangfeng, and Dahuang. ^c^ Chinese medicine, including Huangqi, Danggui, Dangshen, Baishao, Chishao, Taoren, Honghua, Kushen, Tufuling, Difuzi, Baixianpi, Fuling, Dahuang, and Gancao.

## Data Availability

The data used to support the findings of this study are included within the article or Appendix A.

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
