# Peer review of "Clinical Efficacy and Safety of Chinese Herbal Medicine in the Treatment of Uremic Pruritus: A Meta-Analysis of Randomized Controlled Trials"

_pharmaceuticals, 2022, doi:10.3390/ph15101239_

Round 1
Reviewer 1 Report
This manuscript is a meta analysis of the impact of CHM on the severity of UP, and the associated inflammatory and metabolic consequences of this treatment.
While the results of the meta analysis are positive, improvement in pruritus based on VAS reduction is modest (about 2). Effective treatments for pruritus usually will demonstrate a reduction in VAS of 3 or more. Since the placebo responses are high in pruritus studies, modest improvements must be assessed with caution. It is also difficult to have appropriate controls in CHM trials.
Independent of the CHM type of treatment, all products showed very similar results. The authors fail to explain why this is possible. The different products have different components with different mechanisms of action, yet they all work to the same degree. This suggests that the effect is mild, and perhaps independent of the exact product tested, or the mechanism of action of all the products is the same and as yet not identified.
The improvement of renal function purported to have been observed in these trials is a major confounder. Improving renal function alone can reduce uremic pruritus (UP) (ie in hemodialysis patients more aggressive dialysis results in significant reduction of UP). A truly effective agent should be able to reduce the severity of UP without improvement of renal function.
Overall this manuscript represents a significant review on a topic of intense importance, UP. It shows that CHM of various types may modestly improve UP. It does not help the clinician in selecting the best CHM concoction for their patient with UP. It does not provide insight into the pathogenesis of UP. It also does not integrate CHM use into a treatment algorithm with respect of other treatment approaches to UP. Specifically does CHM interact with Western medications used to treat UP? The review by Ko-Lin Kuo of 2021 is useful in that regard.
Author Response
Pharmaceuticals Editorial Team
Manuscript ID: 1845734
TITLE: Clinical Efficacy and Safety of Chinese herbal medicine in the Treatment of Uremic Pruritus: A Meta-Analysis of Randomized Controlled Trials
Dear Editors and reviewers:
Enclosed you will find our revised manuscript entitled " Clinical Efficacy and Safety of Chinese herbal medicine in the Treatment of Uremic Pruritus: A Meta-Analysis of Randomized Controlled Trials " (ID: 1845734). We are pleased that the manuscript was kindly reviewed by the referees and editors, and have made revisions according to the constructive criticisms and suggestions of all reviewers and editors. We would like to thank all of the editors and reviewers for the comments. As request, we have answered all the comments point-by-point as possible as we can. To concentrate on changes in the new manuscript, we highlight the changes with blue text, showing the new materials to provide a more comprehensive understanding of this work. We hope the efforts we did can help to substantiate our results and hope you can consider our revised manuscript for publication in your journal. Our responses to the specific comments are as follows.
Po-Hsuan Lu, MD, PhD
Department of Medicine, Mackay Medical College, New Taipei City, Taiwan
Department of Dermatology, MacKay Memorial Hospital, Taipei, Taiwan
Tel: 886-2-5433535#2556
Fax: 886-2-25232448
Email: pohsuan@gmail.com
Response to reviewer
Reviewer 1
Comments and Suggestions for Authors
This manuscript is a meta-analysis of the impact of CHM on the severity of UP, and the associated inflammatory and metabolic consequences of this treatment.
Suggestion 1: While the results of the meta-analysis are positive, improvement in pruritus based on VAS reduction is modest (about 2). Effective treatments for pruritus usually will demonstrate a reduction in VAS of 3 or more. Since the placebo responses are high in pruritus studies, modest improvements must be assessed with caution. It is also difficult to have appropriate controls in CHM trials.
Response 1: Thank you very much for your insightful comments. Improvement of VAS is modest to minor (Fig 3, 4), although statistically significant after pooling to meta-analyze the expanded study population.
We’ve addressed the modest improvement with caution and discussed the limitation of lack of appropriate controls in CHM trials in our discussion (Page 20, Line 374-375).
Please see our revision:
Forth, inconsistent symptomatic treatments and lack of appropriate controls might lead to the modest reduction of VAS score in the CHM. (Page 20, Line 374-375)
Comment 2: Independent of the CHM type of treatment, all products showed very similar results. The authors fail to explain why this is possible. The different products have different components with different mechanisms of action, yet they all work to the same degree. This suggests that the effect is mild, and perhaps independent of the exact product tested, or the mechanism of action of all the products is the same and as yet not identified.
Response 2: Thank you very much for your suggestions.
Included CHM formulas show different components of herbs (Please see our Table S2 Components of Chinese herbal medicine in the included studies) and different treatment efficacy (Please see our Table 1. Characteristics of selected studies).
We listed different mechanism and effects of CHM formulas for UP as below:
UCG
UCG showed benefits in reducing VAS of UP from 7.21(1.72)→ 1.47(0.34) based on study from Xi et al [1]. The prescription for UCG in TCM includes 16 herbs, such as Danshen, Gancao, Huangqi, Fuling, Kushen, Juhua, Banxia, Baizhu, Fuling, Heshouwu, Chaihu, Cheqiancao, Dahuang, Sangbaipi, Baishao, Chuanxiong. These compounds show synergistic effects to improve renal function, including lowering blood urea nitrogen (BUN) and serum creatinine (Scr) levels [2], balancing the Ca and P metabolic dysfunction [3] and improving the systemic microinflammatory state [4].
Please see Table 1. Characteristics of selected studies
Xi (2021) [1] |
High-flux HD |
UCG+High-flux HD |
58/58 |
I: 47.88(3.52) |
5g, 4 times/d |
NA |
VAS |
I: 7.21(1.72)→ 1.47(0.34) |
Siwu Tang
Siwu Tang showed benefits in reducing VAS of UP from 6.23(1.22)→ 3.33(1.42) based on study from Zhao et al [5]. The prescription for Siwu Tang in TCM includes four herbs, such as Shoudihuang, Danggui, Baishao and Chuanxiong. Siwu Tang attenuated uric acid blood urea nitrogen (BUN) and serum creatinine (Scr) levels in the animal model [6]. Topical application of Danggui alleviated dermatitis by reducing cytokines, such as IL-4, IL-6, TNF-α, and IFN-γ [7]. Baishao exhibited anti-pruritic effects by inhibiting histamine release from mast cells and attenuating IgE-related itching sensation [8,9]. As an effective agents for UP, the effect of herbs in Siwu Tang is different from the effect of herbs in UCG.
Please see Table 1. Characteristics of selected studies
Zhao (2020) [5] |
HD |
Siwu Decoction +HD |
30/30 |
I: 61.17(13.35) |
NA, 2 times/d |
1M |
VAS |
I: 6.23(1.22)→ 3.33(1.42) |
Modified Yangxue Runfu Yin
Modified Yangxue Runfu Yin showed benefits in reducing VAS of UP from 8.02(2.26)→ 3.88(1.84) based on study from Dou et al [10]. The prescription for Modified Yangxue Runfu Yin in TCM includes eleven herbs, such as Huangqi, Danggui, Dihuang, Difuzi, Huangqin, Tianmendong, Maidong, Taoren, Honghua, Tianhuafen and Shengma. Modified Yangxue Runfu Yin reduced serum level of phosphate (P), C-reactive protein (CRP), interleukin (IL)-6, tumor necrosis factor (TNF)-α in dialysis patients according to previous studies [10]. Fructus Kochiae (Difuzi) showed anti-inflammatory effect on dermatitis via pERK1/2/TLR4/NF- κ B pathway [11]. Different CHM formula showed anti-inflammatory effect of UP, however, pharmacological pathway and inflammatory biomarker suppression are different in formula and components.
Please see Table 1. Characteristics of selected studies
Dou (2021) [7] |
Desloratadine +HD |
Modified Yangxue Runfu Yin +Desloratadine +HD |
40/40 |
I: 53.56(15.67) |
100ml, 2 times/d |
0.5M |
VAS |
I: 8.02(2.26)→ 3.88(1.84) |
Reference:
[1] Xi, M.M. Clinical observation of niaoduqing granules combined with high-flux hemodialysis in treating uremic skin pruritus. Guide of China Medicine 2021, 19, 112-113.
[2] Zheng Y., Cai G.-Y., He L.-Q., Lin H.-L., Cheng X.-H., Wang N.-S., Jian G.-H., Liu X.-S., Liu Y.-N., Ni Z.-H. Efficacy and safety of Niaoduqing particles for delaying moderate-to-severe renal dysfunction: A randomized, double-blind, placebo-controlled, multicenter clinical study. Chin. Med. J. 2017;130:2402.
[3] Ye M.-Y., Zheng J., Zhang J. The Influence of Niaoduqing Particle on the Calcium and Phosphorus Metabolism and FGF23 in Patients with Chronic Kidney Disease. Med. Innov. China. 2015;23:16–18.
[4] Yin X.-F., Han L.-L. Effect of uremic clearance granule on the systemic micro-inflammatory state of patients after peritoneal dialysis. Pract. Pharm. Clin. Remedies. 2013;2:125–126.
[5] Zhao, Y. Effect of shenhuangliangyue lotion combined with siwu decoction on skin pruritus of maintenance hemodialysis patients (master thesis). Hunan University of Chinese medicine, Hunan, China, 2020.
[6] Wang R, Ma CH, Zhou F, Kong LD. Siwu decoction attenuates oxonate-induced hyperuricemia and kidney inflammation in mice. Chinese journal of natural medicines. 2016;14(7):499-507.
[7] Lee, J.; Choi, Y.Y.; Kim, M.H.; Han, J.M.; Lee, J.E.; Kim, E.H.; Hong, J.; Kim, J.;
Yang, W.M. Topical application of angelica sinensis improves pruritus and skin
inflammation in mice with atopic dermatitis-like symptoms. Journal of medicinal food
2016, 19, 98-105.
[8] Lee, B.; Shin, Y.W.; Bae, E.A.; Han, S.J.; Kim, J.S.; Kang, S.S.; Kim, D.H. Antiallergic effect of the root of paeonia lactiflora and its constituents paeoniflorin and paeonol. Archives of pharmacal research 2008, 31, 445-450.
[9] Dai, Y.; But, P.P.; Chan, Y.P.; Matsuda, H.; Kubo, M. Antipruritic and antiinflammatory effects of aqueous extract from si-wu-tang. Biological & pharmaceutical bulletin 2002, 25, 1175-1178
[10] Dou LY. Efficacy of Modified Yangxue Runfu Yin in hemodialysis patients with Xue Xu Feng Zao type of uremic pruritus Modern Medicine and Health Research. 2021;5(15):23-5
[11] Xiao Z, Xiao S, Zhang Y, Pan T, Ouyang B. The Anti-Inflammatory Effect of Fructus Kochiae on Allergic Contact Dermatitis Rats via pERK1/2/TLR4/NF-κB Pathway Activation. Evid Based Complement Alternat Med. 2018 Jan 4
Comment 3: The improvement of renal function purported to have been observed in these trials is a major confounder. Improving renal function alone can reduce uremic pruritus (UP) (ie in hemodialysis patients more aggressive dialysis results in significant reduction of UP). A truly effective agent should be able to reduce the severity of UP without improvement of renal function.
Response 3: Thank you very much for your suggestions.
Recent study revealed that microinflammation on skin as a pathologically relevant factor in UP. Stimuli including uremic toxins, histamine, cytokine, parathormone might relate to neuroinflammation and cause itching sensation (see Page 18, Line 260-263) [1].
Regarding effective agent and proposed treatments (figure 2) for UP [2], Omega-3 and charcoal were listed.
Omega-3 alleviated UP in the clinical trial [2] and significantly decreased serum level of CRP and hs-CRP in dialysis patients [3]. According to previous review, n-3 PUFA supplementation reduced cardiovascular mortality of hemodialysis patients [4]. Omega-3 fatty acid was beneficial to kidney dysfunction induced by reperfusion injury in rat models. Decreased serum creatinine and uric acid level was found after Omega-3 fatty acid given [5].
Indoxyl sulfate, initially identified as a major uremic toxin that causes uremic symptoms, contributes to CKD progression. AST-120, a type of charcoal, could absorb indole and reduce its conversion into Indoxyl sulfate to decrease uremic symptoms [6]. In addition, AST-120 administration significantly suppressed eGFR decline rate in moderate to severe CKD patients [7].
However, in aforementioned studies, the effective agents Omega-3 and AST-120 for UP might show renal protection simultaneously. The similar conclusion is also found in CHM for UP according to the result of our meta-analysis. CHM is an effective complementary treatments for UP patient with decreasing inflammatory response, attenuating VAS score and improving renal function (decreased serum creatinine and increased eGFR).
Reference:
[1] Mettang T, Kremer AE. Uremic pruritus. Kidney International. 2015;87(4):685-91.
[2] Lu, P.H.; Tai, Y.C.; Yu, M.C.; Lin, I.H.; Kuo, K.L. Western and complementary alternative medicine treatment of uremic pruritus: A literature review. Tzu chi medical journal 2021, 33, 350-358.
[3] Dezfouli M, Moeinzadeh F, Taheri S, Feizi A. The Effect of Omega-3 Supplementation on Serum Levels of Inflammatory Biomarkers and Albumin in Hemodialysis Patients: A Systematic Review and Meta-analysis. Journal of renal nutrition : the official journal of the Council on Renal Nutrition of the National Kidney Foundation. 2020;30(3):182-8.
[4] Saglimbene VM, Wong G, van Zwieten A, Palmer SC, Ruospo M, Natale P, et al. Effects of omega-3 polyunsaturated fatty acid intake in patients with chronic kidney disease: Systematic review and meta-analysis of randomized controlled trials. Clinical Nutrition. 2020;39(2):358-68
[5] Ashtiyani SC, Najafi H, Kabirinia K, Vahedi E, Jamebozorky L. Oral omega-3 fatty acid for reduction of kidney dysfunction induced by reperfusion injury in rats. Iranian journal of kidney diseases. 2012;6(4):275-83
[6] Asai M, Kumakura S, Kikuchi M. Review of the efficacy of AST-120 (KREMEZIN(®)) on renal function in chronic kidney disease patients. Renal failure. 2019;41(1):47-56.
[7] Akizawa T., Asano Y., Morita S., Wakita T., Onishi Y., Fukuhara S., Gejyo F., Matsuo S., Yorioka N., Kurokawa K. Effect of a carbonaceous oral adsorbent on the progression of CKD: A multicenter, randomized, controlled trial. Am. J. Kidney Dis. 2009;54:459–467.
Comment 4: Overall this manuscript represents a significant review on a topic of intense importance, UP. It shows that CHM of various types may modestly improve UP. It does not help the clinician in selecting the best CHM concoction for their patient with UP. It does not provide insight into the pathogenesis of UP. It also does not integrate CHM use into a treatment algorithm with respect of other treatment approaches to UP. Specifically does CHM interact with Western medications used to treat UP? The review by Ko-Lin Kuo of 2021 is useful in that regard.
Response 4: Thank you very much for your suggestions. We will add the limitations of this study. We will add review by Ko-Lin Kuo 2021 in our references.
According to an updated guideline of The PRISMA 2020 statement, meta-analysis of effect estimates shows a statistical technique used to synthesize results when study effect estimates and their variances are available, yielding a quantitative summary of results [1]. We analyzed and synthesized different outcome measure including effectiveness, VAS, laboratory data and adverse events. However, the choice of CHM concoction should be followed by head-to-head comparison of the CHM efficacy such as methods of network meta-analysis including direct and indirect comparison.
Ko-Lin Kuo [2] reviewed several CHM treatments for UP including Chinese herbal bath therapy, Chinese herbal-based cream and Chinese herbal formula Xiao Feng San. The efficacy of CHM treatments for UP and the comparison between Western medicine and Complementary alternative medicine were well-presented. However, less information of interaction between CHM and Western medications was reported in the study. The recent evidence raises the importance of the adjunctive treatments for refractory UP, the drug interaction between CHM and Western medication should be further studied and verified.
Regarding the mechanism and pathogenesis of UP, a literature review proposed possible mechanism of UP including central stimulus from opioid receptor, deposited toxins and systemic inflammation associated with histamine and pro-inflammatory cytokines such as CRP and IL-6. Treatment of UP is various, including anticonvulsants, opioid receptor agonist, antihistamine or other complementary alternative medicine. However, as recent therapeutic algorithm, complementary alternative medicine was considered as adjunctive treatment while refractory status after systemic treatments such as anticonvulsants, opioid receptor agonist or antihistamine [2].
Please see our revision:
A literature review proposed possible mechanism of UP including central stimulus from opioid receptor, deposited toxins and systemic inflammation associated with histamine and pro-inflammatory cytokines such as CRP and IL-6 [14].” (Page 18, Line 260-263)
As recent therapeutic algorithm, complementary alternative medicine was considered as adjunctive treatment while refractory status after systemic treatments such as anticonvulsants, opioid receptor agonist or antihistamine [14]” (Page 2, Line 60-62)
Reference:
[1] M.J. Page, J.E. McKenzie, P.M. Bossuyt, I. Boutron, T.C. Hoffmann, C.D. Mulrow, L. Shamseer, J.M. Tetzlaff, E.A. Akl, S.E. Brennan, R. Chou, J. Glanville, J.M. Grimshaw, A. Hróbjartsson, M.M. Lalu, T. Li, E.W. Loder, E. Mayo-Wilson, S. McDonald, L.A. McGuinness, L.A. Stewart, J. Thomas, A.C. Tricco, V.A. Welch, P. Whiting, D. Moher, The PRISMA 2020 statement: an updated guideline for reporting systematic reviews, BMJ (Clinical research ed.) 372 (2021) n71.
[2] Lu, P.H.; Tai, Y.C.; Yu, M.C.; Lin, I.H.; Kuo, K.L. Western and complementary alternative medicine treatment of uremic pruritus: A literature review. Tzu chi medical journal 2021, 33, 350-358.

Reviewer 2 Report
Dear authors, thank you for the huge amount of work. I have one comment:
The confidence intervals between patients and controls overlap. For the controls it passes 1 and can not be described as having any meaning. The patients' 95% CI is part (contained within the values) of the controls' CI.
You do not describe the controls. In the abstract and conclusions it is not clear if controls used placebo or other medication. You give different CIs in the results from the one in the abstract conclusion.
Please rephrase the main conclusion. If there is no significant difference this may be because of different factors like the studies being too different, or other. This could be described in limitations, but not conclude with significance as done in the abstract.
Minor comment:
Usually quality of life relighted to a disease is denoted as Health Related Quality of Life (HRQoL) since QoL may mean impairment caused by other factors, not health related, e.g. climate, work, etc
Author Response
Pharmaceuticals Editorial Team
Manuscript ID: 1845734
TITLE: Clinical Efficacy and Safety of Chinese herbal medicine in the Treatment of Uremic Pruritus: A Meta-Analysis of Randomized Controlled Trials
Dear Editors and reviewers:
Enclosed you will find our revised manuscript entitled " Clinical Efficacy and Safety of Chinese herbal medicine in the Treatment of Uremic Pruritus: A Meta-Analysis of Randomized Controlled Trials " (ID: 1845734). We are pleased that the manuscript was kindly reviewed by the referees and editors, and have made revisions according to the constructive criticisms and suggestions of all reviewers and editors. We would like to thank all of the editors and reviewers for the comments. As request, we have answered all the comments point-by-point as possible as we can. To concentrate on changes in the new manuscript, we highlight the changes with blue text, showing the new materials to provide a more comprehensive understanding of this work. We hope the efforts we did can help to substantiate our results and hope you can consider our revised manuscript for publication in your journal. Our responses to the specific comments are as follows.
Po-Hsuan Lu, MD, PhD
Department of Medicine, Mackay Medical College, New Taipei City, Taiwan
Department of Dermatology, MacKay Memorial Hospital, Taipei, Taiwan
Tel: 886-2-5433535#2556
Fax: 886-2-25232448
Email: pohsuan@gmail.com
Response to reviewer
Reviewer 2
Comments and Suggestions for Authors
Comment 1: The confidence intervals between patients and controls overlap. For the controls it passes 1 and can not be described as having any meaning. The patients' 95% CI is part (contained within the values) of the controls' CI.
Response 1: Thank you very much for your comments.
The mean differences were calculated according to formula [1] and implemented with RevMan software (version 5.4). The dichotomous outcomes such as risk ratios were calculated according to instructions [2]. These effect sizes were then adjusted with inverse variance weight [3] for pooling of all the groups and implement in RevMan. Thus, the compiled datasets showed mean differences and risk ratios as shown in Forest plots (Fig. 3-6), with only 1-3 studies crossing the line x=0 or x=1 and most of the studies were in favor of just one side in each meta-analysis, as calculated with z-score, showing statistical significance at p < 0.05.
However, some of sentences were ambiguous or too arbitrary, so we rephrase the lines:
“Compared to control groups, no significantly higher risk of adverse events in patients taking Chinese herbal medicine (Risk ratio 0.60, 95% CI 0.22 to 1.63).” (Page 1, Line 32-34)
Reference:
[1] https://handbook-5-1.cochrane.org/chapter_16/16_4_6_1_mean_differences.htm
[2] https://handbook-5-1.cochrane.org/chapter_9/table_9_4_a_summary_of_meta_analysis_methods_available_in.htm
[3] https://training.cochrane.org/handbook/current/statistical-methods-revman5
Comment 2: You do not describe the controls. In the abstract and conclusion it is not clear if controls used placebo or other medication. You give different CIs in the results from the one in the abstract conclusion.
Response 2:
Thank you very much for your comments. We added “The management of control group and intervention group was listed as Table 1, most of the patients in control group undergoing hemodialysis. Some dialysis patients underwent high-reflux hemodialysis.” (Page 2, Line 88-90) We have mentioned some additional treatments: “Thirteen trials assessed additional treatments including antihistamine, calamine lotion in control group” (Page 2, Line 90-92).
Please see our revision:
The management of control group and intervention group was listed as Table 1, most of the patients in control group undergoing hemodialysis. Some dialysis patients underwent high-reflux hemodialysis (Page 2, Line 88-90)
According to our result, intervention group Chinese herbal medicine treatment showed benefits in increasing ER and reducing VAS score compared to control groups whether with hemodialysis alone or combined with anti-pruritic treatments. We added the description in the abstract “adjunctive Chinese herbal medicine significantly improved overall effectiveness (Risk ratio 1.29, 95% CI 1.23 to 1.35), quality of life, renal function, reduced pruritis score, and inflammatory biomarkers comparing to control groups with hemodialysis alone or with anti-pruritic treatments.” (Page 1, Line 28-31) and in the conclusion “adjunctive CHM improves quality of life, renal function and attenuates inflammation whereas no statistically difference of adverse drug reaction is found comparing to UP patients only received hemodialysis or with anti-pruritic treatments” (Page 21, Line 434-437)
Please see our revision:
Adjunctive Chinese herbal medicine significantly improved overall effectiveness (Risk ratio 1.29, 95% CI 1.23 to 1.35), quality of life, renal function, reduced pruritis score, and inflammatory biomarkers comparing to control groups with hemodialysis alone or with anti-pruritic treatments (Page 1, Line 28-31)
Adjunctive CHM significantly improves quality of life and renal function while also attenuating inflammation, without serious adverse effects comparing to UP patients only received hemodialysis or with anti-pruritic treatments. (Page 21 Line 434-437)
We have confirmed the same risk ratio and confidence intervals in the results from the one in the abstract conclusion.
In abstract conclusion:
Chinese herbal medicine significantly improved overall effectiveness (Risk ratio 1.29, 95% CI 1.23 to 1.35) (Page 1, Line 28-29)
adverse events in patients taking Chinese herbal medicine (Risk ratio 0.60, 95% CI 0.22 to 1.63). (Page 1, Line 33-34)
In the result:
overall effectiveness was significantly higher in patients receiving CHM than for patients in control groups (RR 1.29, 95% CI 1.23 to 1.35) (Page 13, Line 163-165)
No significant increase in ADRSs was observed in patients after using CHM (RR 0.60, 95% CI 0.22 to 1.63) (Page 17, Line 228-230)
Comment 3: Please rephrase the main conclusion. If there is no significant difference this may be because of different factors like the studies being too different, or other. This could be described in limitations, but not conclude with significance as done in the abstract.
Response 3:
Thank you very much for your comments.
We have mentioned “there were some discrepancies in the interventions of the control groups” which might relate to high heterogeneity of result in the discussion (Page 20, Line 366-367). Regarding no statistically significant difference outcome data, we have mentioned including “changes in K, Ca, iPTH, UA, liver enzymes (ALT, AST), and albumin were not found to be significant between CHM administration and controls.” (Page 17, Line 224-226) and “No significant increase in ADRSs was observed in patients after using CHM (RR 0.60, 95% CI 0.22 to 1.63)” (Page 17, Line 228-230). Although the results of reduction in VAS and event risk ratio have a statistically difference between intervention and control groups, and the effect size is modest to low. Thus, CHM might have a helpful yet minor role in relieving uremic pruritus. Confounders such as improvement of renal function, different concoctions of CHM, and heterogeneity in study groups, shall also be considered.
We rephrased the main conclusion:
This systematic review and meta-analysis demonstrates that CHM, including Touxie-Jiedu-Zhiyang decoctions, UCG, and other decoctions reduce pruritus severity based on overall effectiveness and VAS scores. In addition, adjunctive CHM improves quality of life, renal function and attenuates inflammation whereas no statistically difference of adverse drug reaction is found comparing to UP patients only received he-modialysis or with anti-pruritic treatments. Compared to control groups, CHM in-creases overall effectiveness in both UP patients undergoing dialysis and those not undergoing dialysis. In dialysis patients, CHM alleviates UP and reduces the VAS score over time, especially after more than 12 weeks of use. However, for future research, we recommend examining studies with more patients and higher quality studies that focus on head-to-head comparisons among CHM interventions in UP patients. (Page 21. Line 432-442)
Comment 4: Usually quality of life relighted to a disease is denoted as Health Related Quality of Life (HRQoL) since QoL may mean impairment caused by other factors, not health related, e.g. climate, work, etc
Response 4: Thank you very much for your comments.
According to previous studies, Quality of life (QOL) is mentioned as a significant concept in the field of health and medicine [1]. QoL has been used as an outcome measurement in patients with heart disease [2], chronic obstructive pulmonary disease (COPD) [3], and in skin disease [4]. M.Z. Satti et al [5], Scherer JS et al.[6], Fishbane S et al.[7] and Zhang L et al.[8] used quality of life QOL as outcome measurement of uremic pruritus. Hence, quality of life may be used as an outcome measurement for uremic pruritus.
Reference:
[1] K. Haraldstad, A. Wahl, R. Andenæs, J.R. Andersen, M.H. Andersen, E. Beisland, C.R. Borge, E. Engebretsen, M. Eisemann, L. Halvorsrud, T.A. Hanssen, A. Haugstvedt, T. Haugland, V.A. Johansen, M.H. Larsen, L. Løvereide, B. Løyland, L.G. Kvarme, P. Moons, T.M. Norekvål, L. Ribu, G.E. Rohde, K.H. Urstad, S. Helseth, A systematic review of quality of life research in medicine and health sciences, Quality of life research : an international journal of quality of life aspects of treatment, care and rehabilitation 28(10) (2019) 2641-2650.
[2] Karakurt P, Aşılar RH, Yildirim A, Memiş Ş. Determination of Hopelessness and Quality of Life in Patients with Heart Disease: An Example from Eastern Turkey. J Relig Health. 2018 Dec;57(6):2092-2107
[3] Hao G, Qiu Q, Hou L, Gu F. The Effect of Symptom Clusters and Sleep Disorder on Quality of Life among Patients with Chronic Obstructive Pulmonary Disease. J Healthc Eng. 2021
[4] Guo F, Yu Q, Liu Z, Zhang C, Li P, Xu Y, Zuo Y, Zhang G, Li Y, Liu H. Evaluation of life quality, anxiety, and depression in patients with skin diseases. Medicine (Baltimore). 2020 Oct 30
[5] M.Z. Satti, D. Arshad, H. Javed, A. Shahroz, Z. Tahir, M.M.H. Ahmed, A. Kareem, Uremic Pruritus: Prevalence and Impact on Quality of Life and Depressive Symptoms in Hemodialysis Patients, Cureus 11(7) (2019) e5178.
[6] Scherer JS, Combs SA, Brennan F. Sleep Disorders, Restless Legs Syndrome, and Uremic Pruritus: Diagnosis and Treatment of Common Symptoms in Dialysis Patients. Am J Kidney Dis. 2017 Jan;69(1):117-128
[7] Fishbane S, Mathur V, Germain MJ, Shirazian S, Bhaduri S, Munera C, Spencer RH, Menzaghi F; Trial Investigators. Randomized Controlled Trial of Difelikefalin for Chronic Pruritus in Hemodialysis Patients. Kidney Int Rep. 2020 Jan 28;5(5):600-610.
[8] Zhang L, Li Y, Xiao X, Shi Y, Xu D, Li N, Deng Y. Acupuncture for uremic pruritus: A systematic review and meta-analysis. J Pain Symptom Manage. 2022 Aug 30:S0885-3924(22)00871-5. doi: 10.1016/j.jpainsymman.2022.08.017.

Round 2
Reviewer 1 Report
This manuscript is now improved. It is very long and detailed. The conclusions of the manuscript are sound and based on the presented evidence.
Reviewer 2 Report
The authors have responded adequately to the reviewers' comments.
Please read through for typos.